# Third Early “Booster” Dose Strategy in France of bnt162b2 SARS-CoV-2 Vaccine in Allogeneic Hematopoietic Stem Cell Transplant Recipients Enhances Neutralizing Antibody Responses

**DOI:** 10.3390/v14091928

**Published:** 2022-08-30

**Authors:** Abdelhakim Ahmed-Belkacem, Rabah Redjoul, Rozenn Brillet, Nazim Ahnou, Mathieu Leclerc, Dennis Salomón López-Molina, Alexandre Soulier, Aurélie Gourgeon, Christophe Rodriguez, Sébastien Maury, Jean-Michel Pawlotsky, Slim Fourati

**Affiliations:** 1University Paris Est Créteil, INSERM U955, IMRB, 94010 Creteil, France; 2Haematology Department, Assistance Publique, Hôpitaux de Paris, Henri Mondor Hospital, 94000 Creteil, France; 3Department of Virology, Hôpital Henri Mondor, AP-HP, Université Paris-Est, 94000 Creteil, France

**Keywords:** SARS-CoV-2, variant of concern, COVID-19 vaccine, HSCT immunocompromised patients, neutralizing antibody

## Abstract

Immunocompromised individuals generally fail to mount efficacious immune humoral responses following vaccination. The emergence of SARS-CoV-2 variants of concern has raised the question as to whether levels of anti-spike protein antibodies achieved after two or three doses of the vaccine efficiently protect against breakthrough infection in the context of immune suppression. We used a fluorescence-based neutralization assay to test the sensitivity of SARS-CoV-2 variants (ancestral variant, Beta, Delta, and Omicron BA.1) to the neutralizing response induced by vaccination in highly immunosuppressed allogeneic HSCT recipients, tested after two and three doses of the BNT162b2 vaccine. We show that neutralizing antibody responses to the Beta and Delta variants in most immunocompromised HSCT recipients increased after three vaccine doses up to values similar to those observed in twice-vaccinated healthy adults and were significantly lower against Omicron BA.1. Overall, neutralization titers correlated with the amount of anti-S-RBD antibodies measured by means of enzyme immunoassay, indicating that commercially available assays can be used to quantify the anti-S-RBD antibody response as a reliable surrogate marker of humoral immune protection in both immunocompetent and immunocompromised individuals. Our findings support the recommendation of additional early vaccine doses as a booster of humoral neutralizing activity against emerging variants, in HSCT immunocompromised patients. In the context of Omicron circulation, it further emphasizes the need for reinforcement of preventive measures including the administration of monoclonal antibodies in this high-risk population.

## 1. Introduction

Severe acute respiratory syndrome coronavirus 2 (SARS-CoV-2) is the agent responsible for coronavirus disease 2019 (COVID-19). A two-dose regimen of the BioNTech/Pfizer mRNA BNT162b2 vaccine has been shown to be safe and highly effective in preventing infection and symptomatic COVID-19 [1,2]. Nevertheless, waning immunity in vaccinated individuals [3,4,5,6], together with the emergence of SARS-CoV-2 variants of concern (VOC) has been associated with a risk of breakthrough, possibly severe infection in vulnerable vaccinated populations, such as elderly patients or immunocompromised individuals who may not produce adequate amounts of neutralizing antibodies after the first two doses of the vaccine. These populations are also those most exposed to infection, due to their regular interactions with medical staff and other residents in hospitals and/or nursing homes.

Booster vaccination, with a third dose administered approximately 6 months after the second dose, has been shown to enhance the production of neutralizing antibodies [7,8] while providing longer-lasting protection in healthy individuals [9,10]. Nevertheless, little information is available on the efficacy of an early third dose (1 month after the second dose) vaccine booster in immunocompromised patients. We previously showed that early administration of a third dose of BNT162b2 improved the humoral immunogenicity of the vaccine in recipients of HSCT [11] as reported in patients of solid-organ transplantation [12,13]. The emergence of SARS-CoV-2 variants with higher transmissibility than the original strains, such as variant Delta (B.1.617.2), and/or decreased intrinsic susceptibility to neutralizing antibodies, such as variants Beta (B.1.351) or, more recently, Omicron (B.1.1.529), has raised the question as to whether levels of anti-spike protein antibodies achieved after two or three doses of the vaccine efficiently protect against breakthrough infection in the context of immune suppression. Thus, exploring neutralizing activity against different VOCs as surrogate evidence of vaccine efficacy in such patients is crucial to adapt the timing and number of vaccine doses required to protect this particularly vulnerable population.

In France, a third booster dose of the vaccine has been recommended since early 2021 in immunocompromised patients, including hematopoietic stem cell transplant (HSCT) recipients. The third dose has been administered 28 days after the second dose in patients with a suboptimal response, characterized by the lack of anti-spike seroconversion or a low level of anti-spike antibodies after the second dose, as assessed by automated enzyme immunoassay.

In the present study, we used a fluorescence-based neutralization assay to compare the sensitivity of different infectious SARS-CoV-2 variants, including the ancestral variant, variant Beta, variant Delta, and variant Omicron BA.1, to the neutralizing response induced by vaccination in three cohorts of patients: (i) 26 highly immunosuppressed allogeneic HSCT recipients, tested after two and three doses of the BNT162b2 vaccine; (ii) 22 healthy individuals vaccinated with two doses of the BNT162b2 vaccine (control group 1); and (iii) 20 non-vaccinated convalescent immunocompetent patients who had been infected during the first or the second French epidemic wave in 2020 (control group 2).

## 2. Materials and Methods

### 2.1. Patients

Three groups of patients were included in this retrospective study: (i) The first group (*ImmunoSupp-V*) included allogeneic HSCT recipients vaccinated with the BNT162b2 messenger RNA (mRNA) vaccine (Pfizer-BioNTech) 3 months or more after HSCT. All of them received 3 doses of the vaccine given 1 month apart, with the third dose having been administered in patients considered to have a suboptimal immune humoral response, characterized by anti-spike protein receptor-binding domain (S-RBD) IgG levels < 4160 arbitrary units (AU/mL). This threshold was recommended by the manufacturer and is broadly used for surrogate measurements of vaccine protection. The clinical and biological data were retrospectively collected from their medical charts. (ii) The second group (*ImmunoComp-V*) included healthy immunocompetent healthcare workers having received 2 doses of the BNT162b2 vaccine; (iii) the third group (*ImmunoComp-C*) included convalescent immunocompetent patients who had been infected during the first or the second French epidemic wave in 2020. Their characteristics are shown in Table 1. This anonymous retrospective study protocol followed the ethical guidelines of the declaration of Helsinki. Data collection was declared and approved by the French Committee of Data Protection and Civil Liberties (CNIL), registration number n° 2218612v0.

In the *ImmunoSupp-V* group, serum samples were collected after the 2nd vaccine dose (median: 27 days; range: 20–58 days) and the 3rd vaccine dose (median: 26 days; range: 20–34 days). In the *ImmunoComp-V* group, serum samples were collected after the 2nd vaccine dose (median: 22 days; range: 18–30 days). In the *ImmunoComp-C* group, serum samples were collected 22 days to 3 months (median: 45 days) after the onset of symptoms.

### 2.2. Measurement of Anti-S-RBD IgG Antibody Levels

All sera were analyzed for anti-S-RBD IgG titers with the SARS-CoV-2 IgG Quant II assay on an ARCHITECT device (Abbott, Chicago, IL, USA). The assay is an automated enzyme immunoassay that quantifies anti-S-RBD IgG with a lower limit of detection/quantification of 21 AU/mL and a maximal cutoff of linear quantification of 40,000 AU/mL (analytical measuring interval). Samples containing anti-S-RBD titers higher than 40,000 AU/mL were further diluted to extend the measuring interval. All tests were performed by trained laboratory technicians, according to the manufacturer’s standard procedures.

### 2.3. Cell Lines and Viruses

Vero-E6 cells (ATCC, CRL-1586) were maintained in Dulbecco’s modified Eagle medium (DMEM, ThermoFischer Scientific, Waltham, MA, USA) supplemented with 50 international unit (IU)/mL penicillin, 100 µg/mL streptomycin, 10% fetal bovine serum (FBS), and 0.1 µg/mL fungizone (ThermoFischer Scientific). Calu-3 cells (ATCC, HTB-55) were maintained in the same media supplemented with non-essential amino acids (ThermoFischer Scientific) and 10% sodium bicarbonate (Gibco, ThermoFisher Scientific, Waltham, MA, USA).

SARS-CoV-2 (ancestral variant D614G, variant Beta, and variant Delta) was isolated from nasopharyngeal swabs of symptomatic patients infected during the corresponding French epidemic waves. The variant Omicron prototype (BA.1) was provided by EVAg (UVE/SARS-CoV-2/2021/FR/1514 B.A.1.529, Omicron, sample reference 47184). Variants D614G and beta were amplified by passages in Vero-E6 cells, while variant Delta, which did not result in high viral titer in Vero-E6, was amplified by passages in Calu-3 cells. Viral titers were measured by a standard plaque assay using Vero-E6 cells. All experiments were performed in a biosafety level 3 laboratory. Spike gene sequences of the original clinical sample and expanded viruses were determined and compared to the ancestral Wuhan-Hu-1 reference genome (Accession MN908947). Key spike amino acid changes in the viral isolates are summarized in Appendix A. 

### 2.4. Fluorescence-Based Neutralization Assay

To measure actual neutralizing antibody activity, patients’ sera were heat-inactivated for 1 h at 56 °C and subsequently 4-fold serially diluted from 1:2 to 1:2,048. SARS-CoV-2 viruses (3.6 × 10^5^ TCID 50/mL) were then mixed with diluted sera and incubated for 1 h at 37 °C. The mixture was subsequently added to target cells plated the previous day at 30,000 cells/well in clear-bottom black-walled 96-well culture plates. Four hours later, wells were washed with PBS and 100 µL DMEM supplemented with 2% FBS was added. Twenty hours later, the detection of infected cells was performed by means of immunofluorescent labeling using a primary antibody directed against the SARS-CoV-2 nucleoprotein (GTX135357, Euromedex, Souffelweyersheim, France) followed by labeling with a secondary Alexa fluor antibody (594 nm) (A11037, ThermoFisher Scientific). Total cells were labelled with DAPI. Fluorescence was quantified using multipoint fluorescence intensity detection with top optics on a Varioskan LUX multimode reader, operated with SkanIt Software 6.0. The procedure involves the measurement of the fluorescence intensity within a defined arrangement of 29 points on the bottom of each well. Fluorescence was quantified at both 450 nm (DAPI) and 620 nm (Nucleoprotein) for each of the 29 points. Fluorescence at 620 nm was normalized with the corresponding DAPI fluorescence.

### 2.5. Statistical Analyses

The experiments were performed in triplicate. Data are expressed as mean ± SEM or percentages. Statistical differences between the means of two datasets were assessed using the unpaired, two-sided Student’s *t* test. Correlations between the two datasets were calculated using Spearman’s correlation coefficient with Graphpad Prism software, version is 9.4.1, San Diego, CA, USA.

## 3. Results

### 3.1. Effectiveness of a Third “Booster” Vaccine Dose in HSCT Recipients vs. Two Doses Only in Immunocompetent Patients

The 50% neutralization effectiveness (NT_50_) of sera from allogeneic HSCT recipients (*ImmunoSupp-V* group) who received two or three BNT162b2 vaccine doses was assessed by means of a fluorescence-based neutralization assay on different SARS-CoV-2 variants, including the ancestral variant (i.e., harboring D614G), variant beta, variant delta and variant Omicron BA.1 (Figure 1). For each of these variants, the neutralization capacities after two and three doses were compared to those obtained after two BNT162b2 doses in immunocompetent healthcare workers (*ImmunoComp-V* group) and, additionally for the ancestral variant (i.e., harboring D614G), variant beta, and variant delta, in convalescent immunocompetent patients naturally infected with SARS-CoV-2 (*ImmunoComp-C* group), serving as control groups (Figure 2).

As shown in Figure 2 and Figure 3, serum-neutralizing titers increased in the majority of allogeneic HSCT recipients (88%, 92%, 80%, and 73% against variants D614G, beta, delta, and Omicron, respectively) after the third dose, as compared to their response after the second dose. Quantitatively, the increase in neutralizing capacity was significant: 3.8-fold, 3.5-fold, 2.8-fold, and 2.3-fold against variants D614G (*p* < 0.01), beta (*p* < 0.001), delta (*p* < 0.001), and omicron (*p* < 0.001), respectively (Figure 2 and Figure 3). Overall, the proportion of non-responders, defined by an NT_50_ > 0.5 (i.e., the smallest serum dilution) was smaller after the third dose than after only two doses (11% vs. 42% for D614G, 4% vs. 30% for variant Beta, 19% vs. 58% for variant delta, and 42% vs. 73% for variant Omicron, respectively).

When comparing neutralizing capacities between allogeneic HSCT recipients and the two immunocompetent groups, neutralizing capacities were significantly lower with all variants after two doses of the BNT162b2 vaccine in the former than in the two latter groups. In contrast, after three BNT162b2 vaccine doses, serum-neutralizing capacities were not significantly different from those in two-dose-vaccinated healthy individuals for the tested variants. However, the neutralizing capacity was reduced against Omicron BA.1 in all groups including in the three-dose immunocompromised group (*p* = 0.02) (Figure 2).

### 3.2. Relationship between Anti-S-RBD Antibody Responses and Neutralization Titers

Anti-S-RBD antibodies were quantified in the sera from all vaccinated individuals included in the study and plotted against their respective NT_50_ against D614G, variant beta, and variant delta (Figure 4). Spearman correlation coefficients ranged from 0.70 to 0.93 in immunocompromised allogeneic HSCT recipients and from 0.69 to 0.88 in immunocompetent vaccinated individuals. Thus, the correlation was strong between the anti-S-RBD antibody detected in serum and the in vitro neutralizing capacity for these variants, irrespective of the immunological status of the individuals tested.

## 4. Discussion

COVID-19 RNA vaccines prevent infection in the majority of vaccine recipients. However, several SARS-CoV-2 variants of concern, including variants Beta (B.1.351), Delta (B.1.617.2), and more recently, Omicron, have been reported to have the potential to escape vaccine-induced immune responses in individuals with insufficient levels of neutralizing antibodies [14,15,16,17]. In addition, certain groups of patients, in particular immunocompromised individuals, generally fail to mount efficacious immune responses following vaccination [11,18].

Inducing the production of SARS-CoV-2 neutralizing antibodies directed against the spike protein is one of the vaccine’s goals that achieves protection against the virus. In France, national guidelines recommend the use of anti-SARS-CoV-2 spike protein receptor-binding domain antibody titration to establish whether immunocompromised patients are well, poorly, or not at all protected against infection. This biomarker can be detected and quantified by means of commercially available automated enzyme immunoassays that quantify anti-spike antibodies based on the calibration of neutralizing activity against variants that circulated early during the pandemic, i.e., the original Wuhan strain that preceded VOC emergence. However, very few data have been generated on the ability of such antibodies to neutralize VOCs, particularly in immunocompromised patients.

In the present study, we used an in-house neutralization assay to measure the magnitude of neutralization associated with the anti-spike antibody response against several VOCs (including variants Beta, Delta, and Omicron BA.1) in immunocompromised allogeneic HSCT recipients who received two and three doses of the mRNA vaccine. Antibody responses in healthy immunocompetent vaccinated individuals and in non-vaccinated convalescent patients were used as comparators. We show that immunocompromised patients respond suboptimally to two doses of vaccines, but this insufficiency can be compensated by administering a third “booster” dose one month later. Indeed, in our experiments, the neutralization capacity of our HSCT recipients’ sera against variants Beta, Delta, and Omicron increased after three vaccine doses up to values similar to those observed in the two control groups. However, the neutralizing activity against Omicron was still weak in these groups, as has been reported elsewhere [19]. Interestingly, the neutralizing capacity of the sera correlated with the amount of anti-S-RBD antibodies measured by means of enzyme immunoassay.

Our findings have several potential practical implications: (i) Commercially available assays can reliably be used to quantify the anti-S-RBD antibody response in serum, as a reliable surrogate marker of humoral immune protection against circulating VOCs in both immunocompetent and immunocompromised individuals; (ii) increasing anti-S-RBD antibody titers generated after the third vaccine dose in immunocompromised patients correlate with higher neutralizing potency against variants Beta and Delta but an insufficient response against Omicron. In the context of Omicron circulation, these results further emphasize the need for reinforcement of preventive measures including the administration of active monoclonal antibodies in this high-risk population.

## 5. Conclusions

Together, these findings support the recommendation of early vaccine doses as boosters of humoral neutralizing activity against emerging variants, in particular in immunocompromised patients. In the context of Omicron circulation, these results further emphasize the need for reinforcement of preventive measures including the administration of active monoclonal antibodies in this high-risk population.

## Figures and Tables

**Figure 1 viruses-14-01928-f001:**
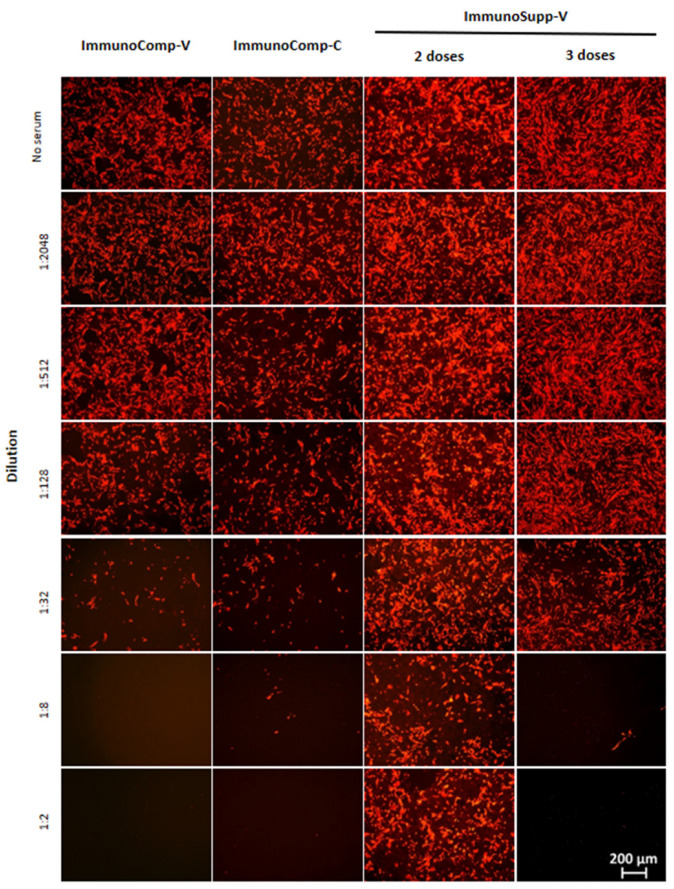
Immunofluorescence labeling of SARS-CoV-2 variant delta infection in Calu-3 cells and neutralization by patient serum dilutions. Representative examples of individuals from the three groups are shown, including *ImmunoSupp-V* (double- then triple-vaccinated allogeneic HSCT recipients), *ImmunoComp-V* (double-vaccinated immunocompetent healthcare workers), and *ImmunoComp-C* (immunocompetent non-vaccinated convalescent individuals).

**Figure 2 viruses-14-01928-f002:**
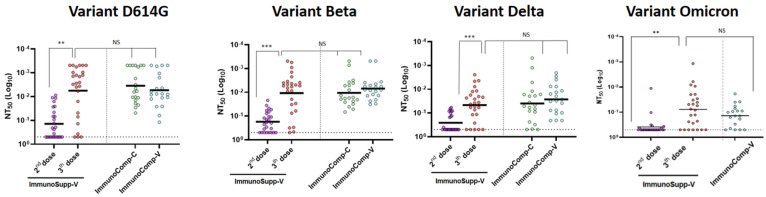
Antibody-mediated neutralization effectiveness of sera from the three groups tested against SARS-CoV-2 variants D614G, Beta, Delta, and Omicron (BA.1). NT_50_ represents the serum dilution resulting in 50% virus neutralization. Neutralization assay was performed using serum samples obtained from double- or triple-BNT162b2 vaccinated immunocompromised allogeneic HSCT recipients (*ImmunoSupp-V*), non-vaccinated immunocompetent convalescent individuals (*ImmunoComp-C*), and double-BNT162b2-vaccinated healthy immunocompetent individuals (*ImmunoComp-V*). Negative titers were handled as 0.5. Statistical significance was calculated by two-tailed, paired Student’s *t* tests. Asterisks indicate *p*-values as **: *p* < 0.01, and ***: *p* < 0.001. NS: Not significantly different.

**Figure 3 viruses-14-01928-f003:**
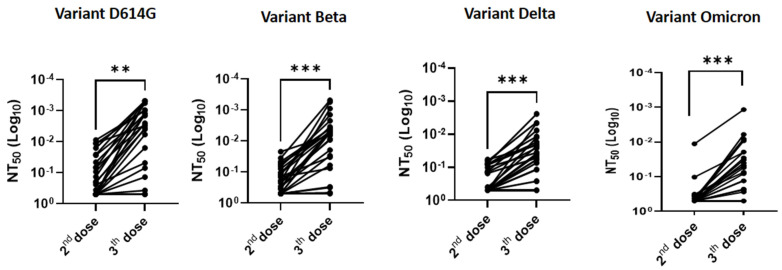
Changes in serum-neutralizing effectiveness against variants D614G, beta, delta, and Omicron (BA.1) after a third dose of BNT162b vaccine compared to post-second vaccine dose in allogeneic HSCT recipients from the *ImmunoSupp-V* group. The experiments were performed in triplicate. Asterisks indicate *p*-values as **: *p* < 0.01 and ***: *p* < 0.001.

**Figure 4 viruses-14-01928-f004:**
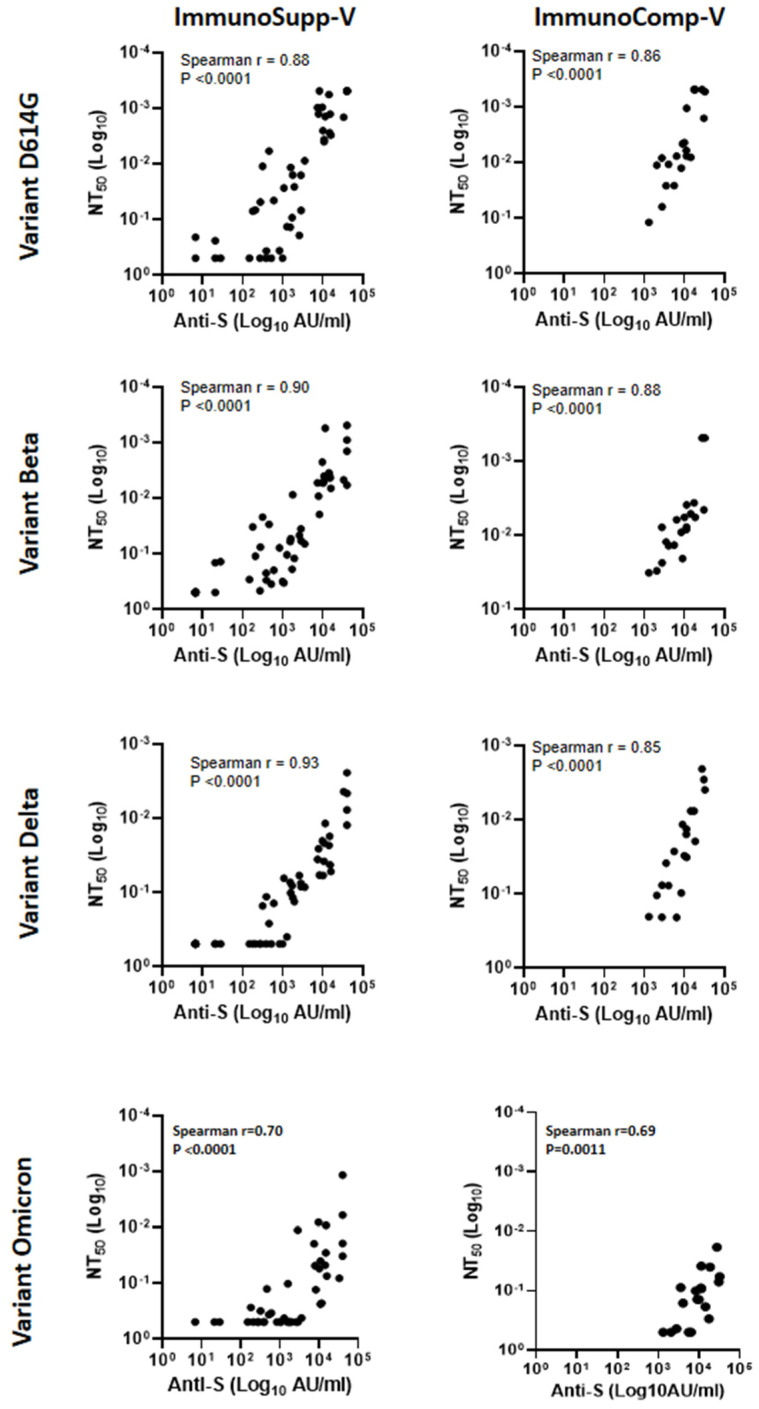
Relationship between serum neutralization titers in cell culture and the amounts of anti-S-RBD antibodies detected in the same sera from individuals in the *ImmunoSupp-V* (double- or triple-BNT162b2 vaccinated immunocompromised allogeneic HSCT recipients) and *ImmunoComp-V* (double-BNT162b2 vaccinated healthy immunocompetent individuals) groups. Anti-S-RBD titers were plotted against the corresponding NT_50_ for variants D614G, Beta, Delta, and Omicron BA.1. Correlations between NT_50_ and anti-S-RBD titers were calculated using Spearman’s correlation, and *p*-values are indicated for each graph.

**Table 1 viruses-14-01928-t001:** Characteristics of the study population, including 3 cohorts: (i) 26 immunocompromised allogeneic HSCT recipients sampled after 2 and 3 doses of BNT162b2 vaccine (*ImmunoSupp-V*); (ii) 22 healthy immunocompetent healthcare workers sampled after 2 doses of BNT162b2 vaccine; (iii) 20 convalescent immunocompetent patients sampled after having been infected during the first or second French epidemic waves in 2020.

	*“ImmunoSupp-V”* Allogeneic HSCT Recipients (n = 26)	*“ImmunoComp-V”* Immunocompetent Vaccinated Individuals (n = 22)	*“ImmunoComp-C”* Immunocompetent Convalescent Patients (n = 20)
Median age (min-max), *years*	61 (31–75)	40 (21–56)	71 (26–94)
% male gender (n/N)	73.1% (19/26)	40.9% (9/22)	40.0% (8/20)
Wards of origin			
Hematology unit [% (n/N)]	100.0% (26/26)	-	-
Geriatric wards [% (n/N)]	-	-	45.0% (9/20)
Medical wards [% (n/N)]	-	-	10.0% (2/20)
Outpatients [% (n/N)]	-	-	20.0% (4/20)
Healthcare workers [% (n/N)]	-	100.0% (22/22)	25.0% (5/20)
Median time between HSCT and initiation of vaccination, (min-max), *months*	14 (3–100)	-	-

## Data Availability

The datasets used and/or analysed during the current study available from the corresponding author on reasonable request.

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
