# Peer review of "Third Early “Booster” Dose Strategy in France of bnt162b2 SARS-CoV-2 Vaccine in Allogeneic Hematopoietic Stem Cell Transplant Recipients Enhances Neutralizing Antibody Responses"

_viruses, 2022, doi:10.3390/v14091928_

Round 1
Reviewer 1 Report
Summary:
The COVID-19 pandemic continues despite the availability and distribution of multiple types of vaccine. Particularly vulnerable populations, such as those with immune deficiencies are vulnerable to infection and severe disease. Vaccine take in vulnerable populations is an important research concern. This study uses neutralization and binding titers of clinical sera to highlight the difficulty in immunizing immune-suppressed individuals with a standard prime/boost strategy. After a third dose, binding and neutralizing antibody titers are increased. Additionally, the researchers assert correlation between binding and neutralizing antibodies in support of using antigen based assays in the clinic.
General Comments:
- Use of the term variant of origin is not standard usage and there are similar terms used by the CDC and WHO. Usage of ancestral, prototype, or wild-type may be less confusing and more accurate.
Specific Comments:
Line 131: Elaborate your method for testing neutralization. How much SARS-CoV-2 was added to the diluted patient sera? Did you use overall levels of fluorescence or the number of infected cells to determine the neutralization endpoint?
Line 149-151: This format text should be removed in the final manuscript.
Reviewer 2 Report
The papers deals with a question that is now quite old. Most immunosuppressed patients have received a fourth vaccine dose, and are approaching the fifth. Furthermore, nAb titers were measured against Beta and Delta, while nowadays we have Omicron
Reviewer 3 Report
Overall, the authors present the data on 3rd booster dose for HSCT patients. There is no data relevant to omicron which is a significant weakness and reduces the utility of the data.
Round 2
Reviewer 3 Report
I have no further comments to strengthen the paper
Author Response
We thank the reviewers and editor for their positive comments.
We modified the text of the manuscript according to the editor's suggestion